# Targeting Key Signaling Pathways in Glioblastoma Stem Cells for the Development of Efficient Chemo- and Immunotherapy

**DOI:** 10.3390/ijms232112919

**Published:** 2022-10-26

**Authors:** Laureen P. Helweg, Jonathan Storm, Kaya E. Witte, Wiebke Schulten, Lennart Wrachtrup, Till Janotte, Angelika Kitke, Johannes F. W. Greiner, Cornelius Knabbe, Barbara Kaltschmidt, Matthias Simon, Christian Kaltschmidt

**Affiliations:** 1Department of Cell Biology, University of Bielefeld, Universitätsstrasse 25, 33615 Bielefeld, Germany; 2Forschungsverbund BioMedizin Bielefeld, OWL (FBMB e.V.), Maraweg 21, 33617 Bielefeld, Germany; 3Heart and Diabetes Centre NRW, Institute for Laboratory and Transfusion Medicine, Ruhr-University Bochum, 32545 Bad Oeynhausen, Germany; 4Molecular Neurobiology, Faculty of Biology, Bielefeld University, Universitätsstrasse 25, 33615 Bielefeld, Germany; 5Department of Neurosurgery and Epilepsy Surgery, Protestant Hospital of Bethel Foundation, University Medical School OWL at Bielefeld, Bielefeld University, Campus Bielefeld-Bethel, Burgsteig 13, 33617 Bielefeld, Germany

**Keywords:** glioblastoma multiforme, cancer stem cells, NF-κB, MYC, NK-cells, chemotherapy, immunotherapy

## Abstract

Glioblastoma multiforme (GBM) is the most aggressive and most common malignant brain tumor with poor patient survival despite therapeutic intervention. On the cellular level, GBM comprises a rare population of glioblastoma stem cells (GSCs), driving therapeutic resistance, invasion, and recurrence. GSCs have thus come into the focus of therapeutic strategies, although their targeting remains challenging. In the present study, we took advantage of three GSCs-populations recently established in our lab to investigate key signaling pathways and subsequent therapeutic strategies targeting GSCs. We observed that NF-κB, a crucial transcription factor in GBM progression, was expressed in all CD44^+^/CD133^+^/Nestin^+^-GSC-populations. Exposure to TNFα led to activation of NF-κB-RELA and/or NF-κB-c-REL, depending on the GBM type. GSCs further expressed the proto-oncogene MYC family, with MYC^high^ GSCs being predominantly located in the tumor spheres (“GROW”-state) while NF-κB-RELA^high^ GSCs were migrating out of the sphere (“GO”-state). We efficiently targeted GSCs by the pharmacologic inhibition of NF-κB using PTDC/Bortezomib or inhibition of MYC by KJ-Pyr-9, which significantly reduced GSC-viability, even in comparison to the standard chemotherapeutic drug temozolomide. As an additional cell-therapeutic strategy, we showed that NK cells could kill GSCs. Our findings offer new perspectives for developing efficient patient-specific chemo- and immunotherapy against GBM.

## 1. Introduction

Glioblastoma multiforme (GBM) is the most common malignant brain and central nervous system tumor (49.1%), accounting for 58.4% of all gliomas [1]. Because of its aggressive nature, GBM is a very deadly neoplasm with an average survival of 12–15 months after diagnosis [2]. The current common treatment strategy is maximal safe surgical resection, followed by radiation and adjuvant chemotherapy with temozolomide (TMZ) [3]. GBMs can be classified as primary, which develops rapidly without a less malignant precursor and accounts for most GBM, as well as secondary, where low-grade tumors develop into a GBM over time [4]. Additionally, two prognostic molecular biomarkers are included in GBM typing, namely isocitrate dehydrogenase (IDH) mutations (typically absent in primary GBM [5]) and O6-methylguanine-DNA methyltransferase (MGMT) promoter methylation [6]. Upregulated MGMT is commonly associated with resistance to TMZ [7,8]. On the molecular level, Happold and coworkers demonstrated MGMT-overexpression to result from increased activity of the transcription factors nuclear factor kappa-light-chain enhancer of activated B-cells (NF-κB) [9]. NF-κB is a key player in a range of vital cellular processes, including inflammation and proliferation, while its unregulated activity drives the initiation and progression of cancer [10,11,12], particularly GBM. Here, NF-κB is commonly observed to be constitutively active in GBM in turn, driving, amongst others, GBM invasiveness, angiogenesis, and resistance to radiotherapy (reviewed in [13,14]). Next to NF-κB, MYC transcription factors, including MYC, N-MYC, and L-MYC, are key players in cell growth and brain development and are implicated in most human cancers (reviewed in [15,16,17]). MYC was also reported to be highly expressed in GBM [18,19], and overexpression of MYC in gliomas is related to increasing tumor grade [20]. While the functions of NF-κB and MYC seem to overlap, it was also shown that NF-κB signaling could regulate MYC signaling [21,22,23,24,25,26].

Besides therapy with small-molecule drugs, cellular immunotherapy in tumors is rapidly rising in importance, with more and more cellular products gaining U.S. Food and Drug Administration (FDA) approval [27]. Natural killer (NK) cells, with their innate ability to differentiate between abnormal and healthy cells, are generally accepted as cells with immunotherapeutic potential against tumors, such as GBM [28]. Linking MYC signaling in tumors to NK cells, MYC upregulation was found to reduce the surface expression of NKG2D ligands in K562 cells. The NKG2D receptor is a master regulator of cytotoxic responses in various immune cells, including NK cells, and involved in the recognition of stressed cells [29,30]. Also, Kortlever and coworkers showed in an in vivo adenocarcinoma model that stromal changes and tumor regression, which appeared after the deactivation of MYC, were mediated by NK cells. Additionally, other groups have demonstrated that NF-κB signaling influences the expression level of death receptors and NK cell ligands in neuroblastoma cells or NK cell ligands in senescent cells, as well as proinflammatory cytokine release in triple-negative breast cancer [31,32,33].

In the present study, we assessed the signaling of NF-κB and MYC with potential therapeutic applications in primary cancer stem cells (CSCs) isolated from GBM. On the cellular level, GBM tumors are highly heterogeneous and comprise differentiated and partly differentiated cells, quiescent GBM cells, and glioblastoma stem cells (GSCs). Differentiated and incompletely differentiated cells (including respective progeny) constitute the tumor mass and are highly proliferative [14,34]. On the contrary, self-renewing GSCs are relatively rare in GBM but show tumor initiation capabilities [35]. Unlike their differentiated counterparts, GSCs also drive therapeutic resistance, invasion, angiogenesis, and immune evasion [34,36,37]. In addition, GSCs further promote tumor recurrence due to their localization within the tumor-infiltrating zone of GBM [38]. Targeting CSC as drivers of GBM has thus become one major focus of therapeutic strategies [37,39], emphasizing the need for suitable primary human in vitro models to investigate key signaling pathways and subsequent inhibition strategies.

Facing this challenge, we recently successfully isolated GSCs from three GBM patients and enriched the GSC pool by serial trypsinization as well as exposure to epithelial growth factor (EGF) and basic fibroblast growth factor (bFGF) [40], in accordance with our own and other previous findings [41,42,43]. In the present study, we took advantage of these three recently established GSCs-populations to investigate both NF-κB and MYC-signaling and their therapeutic relevance in GSCs. Our findings reveal that NF-κB- and MYC-signaling are both present in CD44^+^/CD133^+^/Nestin^+^ GSCs, with MYC^high^ GSCs being located in the tumor spheres while NF-κB-RELA^high^ GSCs were outgrowing. As a basis for new therapeutic strategies, we show that pharmacologic inhibition of NF-κB by ammonium pyrrolidine dithiocarbamate (PDTC) or inhibition of MYC/Myc-associated factor X (MAX) interaction by KJ-Pyr-9 results in a significant reduction of GSC-viability, even in comparison to TMZ-treatment. We also show that Bortezomib is even more efficient than PDTC in reducing GSC viability. Next to pharmacologic inhibition of key signaling pathways, immunotherapeutic strategies are increasingly recognized as promising tools for treating GBM [44,45]. Here we demonstrate that NK cells are capable of efficiently killing GSCs, which offers a further perspective for targeting GSCs in addition to MYC- or NF-κB-inhibition.

## 2. Results

### 2.1. Establishment of Primary Glioblastoma Stem Cells

In the present study, we took advantage of the three recently established GSCs populations [40] to investigate both NF-κB and MYC-signaling and their therapeutic relevance. Of the three patients (two male, one female, 42–86 years), the female patient (GII), as well as one male patient (GV), suffered a primary form of GBM with an *IDH1* wildtype, whereas patient two (GIV) owned a secondary GBM and showed a mutation of the *IDH1* gene (Table 1). Donors GII and GIV further revealed characteristic *MGMT* promoter methylations (Table 1).

Prior to GSC isolation and culture (Figure 1A) [40], histopathological analysis revealed the characteristic cancer cell accumulation within the infiltrated brain tissue via hematoxylin/eosin staining (exemplary depicted in Figure 1B). Further, immunohistochemical staining of paraffin-embedded GIV tissue sections revealed the expression of CSC marker CD44/CD133 localized in small nests clustering together (Figure 1C,D). Next, CSC selection was conducted using serial trypsin treatment, resulting in the presence of characteristic spindle-/mesenchymal-shaped GSCs (Figure 1E) [40]. Isolated GSCs could also give rise to tumor spheres (Figure 1F).

To examine stemness properties in GSCs, the protein expression of established CSC markers CD44, CD133, and Nestin was investigated via immunocytochemical staining. It revealed a robust expression of each marker in each cell population culture under adherent conditions (Figure 2A,C and Appendix A) as well as spheres (Figure 2B,D and Appendix A) and in high passages under adherent conditions (Appendix A). Interestingly, CD133 staining appears to be mainly located in the cytoplasm. As a further indication of CSC characteristics, we also analyzed GCS mRNAs via qualitative and quantitative reverse transcriptase polymerase chain reaction (qRT-PCR; Figure 2E,F). Analysis of primary cells on a post-transcriptional level revealed specific expressions for the proto-oncogene *MYC* as well as MYC family member *MYCN* and pluripotency-associated gene octamer-binding transcription factor 4 (*OCT4*) as well as sex-determining region Y-box 2 (*SOX2*). We further observed similar expression levels of *MYC*, *MYCN*, *OCT4,* and *SOX2* in adherent cells and spheres of GII and GIV, showing that culture conditions do not affect stemness marker expression (Figure 2F). Further analysis of the CSC-related aldehyde dehydrogenase (ALDH1) via flow cytometry revealed a respective ALDH1^high^ amount of 6.06% for GII, 26.00% for GIV, and 3.25% for GV (Figure 2G and Appendix A). Taken together, the here detected protein expression of established CSC markers and mRNA expression of stemness-related genes initially confirms the isolation of GSC populations.

### 2.2. Canonical NF-κB Is Expressed in Glioblastoma Stem Cells and Activated by TNFα

As the transcription factor NF-κB is involved in several pathways linking carcinogenesis with inflammation, the protein expression of NF-κB subunits RELA, RELB and c-REL was investigated in isolated GCS populations using immunocytochemical stainings. Each cell population showed expression of all subunits, although expression of RELA was the most dominant (Figure 3A,C and Appendix A). As RELA was predominantly located within the cytoplasm, we aimed to induce nuclear translocation and activation of RELA using stimulation with TNFα. A highly significant induction of RELA was only observable in the GIV cell population (Figure 3B). Thus, nuclear translocation of the other canonical NF-κB subunit c-REL was analyzed after stimulation with TNFα. Here, a significant induction of c-REL was observable after 60 min for GII, GIV, and GV GCSs (Figure 3D), indicating that TNFα activates c-REL in primary GBM-derived stem cells and RELA and cREL in secondary GBM-derived stem cells.

### 2.3. MYC and NF-κB Expression Is Characteristic for Stemness Potential in Glioblastoma Stem Cells

MYC and its MYC-family members are tightly regulated transcription factors involved in various cellular processes like cell proliferation and growth. However, MYC is one of the most deregulated genes in a variety of cancers. Here, we analyzed the haploid copy number of *MYC* and *MYCN* and found a normal copy number for each cell population, although a slightly enhanced copy number of *MYC* and a reduced copy number of *MYCN* was detected for GIV GSCs (Figure 4A). Further analyzing the expression of MYC on protein level via immunocytochemistry revealed a nuclear expression for GII GSCs, a cytosolic expression for GIV GSCs, and a partial nuclear expression for GV GSCs as well as a robust expression of MYC in all cell population-derived spheres (Figure 4B,C and Appendix A). Similarly, cytosolic expression of N-MYC was detected for each cell population, with GII GSCs showing a reduced expression compared to GIV and GV GSCs (Figure 4B,C and Appendix A).

As MYC is a known NF-κB target and TNFα seems to activate canonical NF-κB in the here isolated GSC populations, we aimed to analyze the quantitative expression levels of *MYC* and its family members after TNFα stimulation and/or NF-κB inhibition using PDTC. Here, untreated GSCs generally show higher expression of *MYC* and *MAX* than *MYCN* (Figure 4D), which is consistent with the *MYC/MAX* expression of TNFα/PDTC treated cells (Figure 4E–G). Further, there is no significant difference in expression levels of treated GSCs compared to the control, suggesting that NF-κB does not regulate the expression of *MYC* in GSCs.

As MYC is a transcription factor highly involved in cell growth and RELA has been shown to promote cellular migration, we performed sphere migration assays using GIV GSC-derived spheres followed by immunocytochemical staining against MYC and RELA (Figure 5A). Here, we detected a robust MYC expression within the spheres, indicating the involvement of MYC in sphere growth (Figure 5B). High expression of RELA was detected in the adherent migrating cells, emphasizing the involvement of RELA in GSC cell migration (Figure 5C).

Next, we further analyzed the influence of NF-κB and MYC. We treated GSCs with the NF-κB inhibitor PDTC and the small molecule KJ-Pyr-9, an inhibitor of MYC/NMYC and MAX protein-protein interaction, alone or with TNFα stimulation or co-treatment with standard GBM chemotherapeutic TMZ (Figure 6 and Appendix A). Treatment of each cell population with TNFα, PDTC, TMZ, and KJ-Pyr-9 led to similar survival rates, although GII GSCs showed higher resistance to PDTC and KJ-Pyr-9 but lower resistance to TMZ compared to GIV and GC GSCs (Appendix A). Merging the data of all three subpopulations revealed that stimulation of GSCs using TNFα duplicated their cellular viability, while NF-κB inhibition significantly decreased cellular viability to 11.99% (±2.83; Figure 6A). Co-treatment with TNFα and PDTC led to a similar cellular viability of 11.62 (±1.73), indicating that PDTC reverses the cytoprotective effects of TNFα. Nevertheless, cytoprotective effects of TNFα were observable in GSCs treated with TMZ and/or TNFα, as using TMZ alone significantly decreased cellular viability of GSCs to 64.05% (±10.61) while co-treatment with TNFα significantly enhanced cellular viability to 131.50% (±11.27) (Figure 6B). However, co-treatment of TMZ with TNFα and/or PDTC again significantly decreased cellular viability to 10.55% (±1.90) and 14.59% (±0.85).

Treating GSCs with small molecule KJ-Pyr-9 also significantly decreased cellular viability to 11.83% (±3.00) (Figure 6C). Again, co-treatment with TNFα did not show any cytoprotective effects resulting in cellular viability of 12.90% (±2.72). Surprisingly, co-treatment using PDTC and KJ-Pyr-9 and/or TNFα decreased cellular viabilities to 24.37% (±3.89) and 21.50% (±3.99), respectively, twice as high as KJ-Pyr-9 and PDTC treatment alone. Co-treatment using KJ-Pyr-9 and TMZ resulted in a similar cellular viability as KJ-Pyr-9 alone of 11.15% (±1.75). Similarly, co-treatment using KJ-Pyr-9 and TMZ together with TNFα or PDTC decreased cellular viabilities to 12.72% (±2.73) and 17.05% (±5.03), respectively. Treating GSCs with KJ-Pyr-9, TNFα, PDTC, and TMZ reduced the cellular viability to 25.56% (±5.88), showing that inhibition of MYC using KJ-Pyr-9 alone or inhibition of NF-κB using PDTC solely is the most effective GSC survival-decreasing treatment. Analyzing the action of simultaneous treatment with PDTC and KJ-Pyr-9 using CompuSyn revealed combination index (CI) values greater than 1, and all combinations points were in the antagonistic area within the normalized isobolograms, demonstrating antagonistic actions between PDTC and KJ-Pyr-9 within GII and GIV GSCs (Figure 6D). Interestingly, our IC_50_ value analysis further showed that even very low concentrations of PDTC (1 µM, Appendix A) led to the killing of GSCs, although we applied a concentration of 100 µM PDTC, which is commonly reported in the literature for Jurkat T Cells [46]. Since treatment may reduce stemness of CSCs, the expression of CSC markers CD133, CD44, and Nestin was investigated via quantitative RT-PCR, which showed enhanced expression levels of each marker in GII GSCs as well as similar expression of each marker in GIV GSCs after PDTC and/or KJ-Pyr-9 treatment (Figure 6E). Additionally, expression levels were similar compared to the solvent control.

### 2.4. Inhibition of Proteasome Reduces GSC Survival

The current strategy for NF-κB inhibition in clinical multiple myeloma treatment involves using proteasome inhibitors such as Bortezomib. Here, we analyzed the effect of Bortezomib in various concentrations on GSCs (Figure 7). For each cell population, a concentration above 2.5 nM strongly inhibits cell survival of GSCs (Figure 7A). Calculating the half-maximal inhibitory concentration revealed varying resistances of the cell populations, with GV being the most sensitive, with an IC_50_ value of 2.85 nM (Figure 7B). GIV is the most resistant cell population with an IC_50_ value of 6.69 nM, and GII displayed an IC_50_ value of 3.53 nM.

### 2.5. Natural Killer Cell-Mediated Lysis of Glioblastoma Stem Cells

Given that immunotherapy is an emerging anti-tumor strategy, we next analyzed the effects of NK cell-mediated tumor lysis on primary GSCs. We purified NK cells and cultured them in IL2 (Figure 8A). After incubation of GIV GSCs with NK cells in different ratios (1:1, 1:3, 1:9 target to effector cells) for four hours, flow cytometry was performed to determine the proportion of dead to living target cells. Co-culturing GSCs and NK cells in a ratio of 1:1 resulted in a specific lysis of 6.1% in GII, 19.4% in GIV, and 21.9% in GV, while a higher ratio of 1:3 enhanced the specific lysis to 14.2%, 33.6%, and 35.8%, respective 31.1%, 55.2% and 46.6% in a ratio of 1:9 (Figure 8B). The tendency of GII to be less susceptible was not statistically significant.

## 3. Discussion

Within the present study, we analyzed signaling pathways with potential relevance for cell proliferation, migration, and cell survival in primary glioblastoma stem cells. We confirmed stemness by showing co-expression of established CSC-markers CD133, CD44, and Nestin, as well as sphere formation in GSCs from all three donors. Furthermore, ALDH1 expression was analyzed in all GSC populations, depicting heterogeneous amounts of ALDH1^high^ GSCs. On a mechanistic level, we found TNFα-mediated activation of NF-κB RELA and/or c-REL depending on the tumor type. Next to NF-κB, we analyzed the presence of the proto-oncogene MYC family, revealing an elevated expression of MYC and MAX compared to N-MYC and L-MYC independent of NF-κB activation. Pharmacologic inhibition of NF-κB by PTDC or inhibition of MYC by KJ-Pyr-9 resulted in a significant reduction of GSC viability even in comparison to TMZ treatment. Furthermore, we could demonstrate that the clinically used NF-κB inhibitor Bortezomib efficiently reduced GSCs survival in a single-digit nanomolar range. As an additional cell-therapeutic strategy, we demonstrate here that NK cells are capable of killing GSCs in vitro (Figure 9).

According to our applied criteria, the here analyzed CSCs growing adherently or as spheres fulfill all tested stemness criteria, including the expression of CD44, CD133, and Nestin in accordance with our own and other previous studies [40,47,48]. Importantly, while CD133 was detected as a surface antigen of hematopoietic stem cells and was described as an extracellular GSC marker, Paola Brecia and colleagues demonstrated that GSCs may only express intracellular CD133 without extracellular expression [49,50,51,52]. Within the literature, the so-called magic four transcription factors KLF4, OCT4, SOX2, and MYC were discussed as potential drivers of stemness in CSCs [53]. Here, GSCs expressed MYC, NMYC, and *OCT4* while simultaneously lacking expression of *KLF4*. OCT4 was described as a marker for high-grade GBM and promoted proliferation and colony formation of glioma cells [54]. In our analysis, GSCs from secondary GBM revealed a trend to lower expression of OCT4 compared to GSCs from primary GBM. Despite the very recently discussed role of SOX2 in driving GBM stemness and tumor propagation [55], we found the dimerization partner of OCT4 to be expressed on a low level. In a recent study, a principal component analysis of genes expressed in various cancer stem cells has depicted the here-analyzed secondary GBM as sticking out from all other samples. On the other hand, the primary GBM populations (n = 2) clustered together with CSC cultures from prostate and lung cancer [40]. On the molecular level, we here observed an *IDH1* mutation in the secondary GBM, which was not present in the primary samples, as depicted in Table 1. In this line, *IDH1/2* gene mutations are commonly found in secondary GBM, although this difference seems not to impact patient survival [56] (reviewed in [57,58]). In addition, our findings show a strongly increased amount of ALDH^high^ cells in the GSC population from secondary GBM compared to their counterparts derived from primary GBM. Interestingly, ALDH is discussed in the field as a mediator of resistance to temozolomide [59].

On a mechanistic level, the transcription factor NF-κB is a well-known driver of GBM invasiveness, angiogenesis, and resistance to radiotherapy [13,14]. As we recently reviewed [60], findings by Bredel and coworkers strongly suggest a mutual exclusion of NFKBIA (IκB-α) deletion or EGFR amplification oncogene [61]. Here, the canonical NF-κB subunits RELA and c-REL are highly expressed and localized in the cytoplasm of GSCs. Accordingly, we previously reported the GO term “NF-κB binding” to be enriched in an RNA-Seq analysis of the here investigated GSC populations [40]. Stimulating GSCs with TNFα resulted in a strong and significant nuclear translocation of RELA and cREL in secondary GBM-derived CSCs. On the contrary, GSCs from primary GBM showed a significant nuclear translocation of only c-REL upon exposure to TNFα. This distinct subunit-specificity extends our discussion on RELA and/or c-REL being present in solid cancers [12] by showing the subunit-specificity depending on the GBM tumor type. On the cellular level, TNFα is one of the pro-tumor cytokines enhancing viability in all analyzed GSC populations. Increased survival was observable even upon co-treatment with the standard chemotherapeutic drug TMZ, which is discussed in detail below.

Next to NF-κB, we focused on its target gene MYC, which belongs to the MYC basic helix-loop-helix transcription factor family being highly expressed in GBM [18,19]. Overexpression of MYC in gliomas was further shown to be related to increasing tumor grade [20]. Despite a low expression of *MYC*, *MAX*, *MYCN,* and *MYCL* on the mRNA level, the here isolated GSC populations show high levels of MYC protein. Interestingly, TNFα-dependent stimulation of NF-κB, as well as its simultaneous inhibition by PDTC, showed no effect on the mRNA expression levels of *MYC*, *MAX*, *N-MYC,* and *L-MYC*. Although high MYC protein levels were observable in all GSC populations, only GSCs derived from primary GBM showed predominantly nuclear localization independent of the culture condition. On the contrary, secondary GBM-derived GSCs revealed a predominantly cytosolic localization of MYC. Accordingly, Wang and coworkers demonstrated that MYC is required for regulating the proliferation and survival of glioma CSCs. In detail, small hairpin RNA-mediated knockdown of MYC attenuated proliferation and resulted in a cell cycle arrest in the G0/G1 phase [62]. In this line, MYC was shown to drive self-renewal of CSCs isolated from GBM [63] (reviewed in [64]).

The present study further depicted MYC^high^ GSCs located in the tumor sphere, while NF-κB-RELA^high^ GSCs were migrating out of the sphere. These observations are in accordance with our mRNA expression data showing no changes in *MYC* mRNA levels upon stimulation or inhibition of NF-κB, as discussed above. Our findings suggest MYC^high^ GSCs represent a more proliferative “GROW” subpopulation, while the more invasive “GO” GSC subpopulation consists of NF-κB-RELA^high^ GSCs. This “GO or GROW” hypothesis was already reported for melanoma, proposing transcriptional signature switches of melanoma cells in response to the microenvironment in vitro and in vivo [65,66]. Our present findings indicate these distinct cellular “GO or GROW” states also for GSCs, which is in accordance with a “GO or GROW” transition from proliferative to invasive GBM phenotypes proposed by Hatzikirou and coworkers [67]. In this line, NF-κB was already reported to drive the switch of GSCs to an invasive mesenchymal (“GO”) state correlating with shorter survival in patients suffering from GBM [68] (reviewed in [13]). On the contrary, MYC was shown to regulate the proliferation and cell cycle progression of GSCs [62], as discussed above. These findings emphasize the need for targeting NF-κB and/or MYC in GSCs to inhibit the proliferative and invasive phenotype of GBM.

To further evaluate NF-κB and MYC signaling, we treated GSCs with NF-κB inhibitor PDTC and MYC/MAX inhibitor KJ-Pyr-9. Treatment with PDTC led to a significant reduction of cellular viability, which stands in line with the reported contribution of NF-κB to GSC maintenance. Inhibition of canonical NF-κB activity in patient-derived GSC cultures was reported to drastically decrease tumor-sphere formation frequency [69]. Additionally, PDTC treatment and co-stimulation with TNFα had no cytoprotective effect on GSCs. In contrast to the recently reported GBM viability decreasing and apoptosis enhancing co-treatment of NF-κB inhibitor BAY-7082 and TMZ [70], we did not detect any synergistic effects of PDTC and TMZ treatment. Further, TMZ treatment alone led to a viability reduction of 60%, while PDTC decreased GSC viability to 10%. Interestingly, dual inhibition of NF-kB and MYC resulted in antagonistic actions with a combination index value over one and combination points within the antagonistic area in the normalized isobologram plot. Additionally, no synergistic effect was observed on the dual inhibition of NF-κB and MYC/MAX in GSCs. In contrast, in normal non-transformed s hepatocytes, dual inhibition of NF-kB and MYC showed synergistic actions on cell survival [71].

Similarly, no synergistic effect was reported for patient-derived lung CSCs when dual inhibition of NF-κB and MYC/MAX was applied [48]. Inhibition of MYC/MAX alone significantly reduced cell viability to 12%, which concords with the reported contribution of MYC to glioma stem cell maintenance [62]. Although TMZ was reported to inhibit GBM progression by suppressing MYC [72], we observed a higher viability-decreasing effect of MYC inhibition alone. Further, KJ-Pyr-9 treatment and co-stimulation with TNFα had no cytoprotective effect on GSCs, nor did co-treatment with KJ-Pyr-9, TMZ, and TNFα. Interestingly, co-treatment GSCs with PDTC and KJ-Pyr-9 alone or together with TMZ and TNFα led to higher viability rates than KJ-Pyr-9 alone. Our results indicate that targeting MYC or NF-κB in GSCs are highly promising therapeutic options. Notably, during a cycle of chemotherapy treatment in patients, there are usually around 4–8 cycles of treatment [73]. Therefore, a few surviving CSCs in vivo are thought to be eradicated in patients. PDTC is a member of pharmakons with similar chemical structures, such as DDTC and disulfiram. PDTC is widely used in vitro, but only DDTC and disulfiram are clinically approved drugs. Disulfiram (Antabuse) is used for the treatment of alcoholism, while DDTC is used as an inhibitor of AIDS progression and induces apoptosis in cancer cells [74]. KJ-Pyr-9 is not yet in clinical use, but a proteinous MYC inhibitor (Omomyc, OMO-103) is clinically tested in solid tumors in a phase I/II study (clinical trial number NCT04808362 initiated in 2021). Previously, it was shown that the clinically used NF-κB inhibitor Bortezomib can inhibit GBM cells [75,76] (reviewed for clinical use in [77]). In the GSC populations described here, we show that Bortezomib is already efficient in a single-digit nanomolar dose, with IC_50_ values ranging between 2 and 7 nM.

In addition to the pharmacologic inhibition of NF-κB or MYC, we focused on a cell-therapeutic strategy by utilizing NK cells. Immunotherapy of GBM by NK cells is increasingly gaining interest for therapeutic applications since NK cells possess great cytotoxic activity with less toxic side effects than conventional therapeutic strategies (reviewed in [45]). In a clinical trial, Ishikawa and colleagues reported a reduced tumor volume after treating nine patients diagnosed with malignant glioma with autologous NK cells. Notably, the authors did not observe any signs of severe neurological toxicity [78]. Interestingly, GBM cells with stemness properties seem highly susceptible to NK cell-mediated killing due to lacking expression of MHC class I while simultaneously expressing ligands of activating NK receptors [79,80]. Haspels and coworkers further demonstrated the cytotoxic effect of NK cells against CD133^+^/Nestin^+^-GSCs cultivated as spheres to be more elevated compared to adherently cultivated GSCs lacking expression of CD133 and Nestin [81]. In the present study, we extend these promising findings by showing the expression of CD44, CD133, and Nestin in GSCs independent of their culture condition. Adherently grown CD44^+^CD133^+^/Nestin^+^-GSCs were further killed by NK cells, offering a promising future perspective for targeting GSCs. Friebel and coworkers showed in a single-cell transcriptomic study that CD16-positive NK cells primarily accumulate in *IDH1* mutated gliomas and further in those with negative *MGMT* promoter methylation status [82]. In contrast to GIV, which is *IDH1* mutated, and GV, which is negative for *MGMT* promoter methylation, GII is *IDH1* wildtype and *MGMT* promoter methylated. As peripheral blood NK cells consist primarily of the CD56 dim subset, which is essentially CD16 positive, the tendency for the lower lysis of GII, although not statistically significant, might be explainable [83].

In summary, we used the here-characterized GSCs as in vitro models to study signaling pathways in GBM, showing that NF-κB and MYC/MAX are expressed in CD44^+^/CD133^+^/Nestin^+^-GSC-populations. We efficiently targeted GSCs by pharmacologic inhibition of NF-κB or MYC, which more significantly decreased GSC viability than the standard chemotherapeutic drug TMZ. As a further cell-therapeutic strategy, we showed that NK cells could kill GSCs. Our findings offer new insights for developing potential targeted chemo- and immunotherapy against GBM to overcome CSC-driven treatment failures. This study thereby opens the opportunity to use this system to investigate pharmacological pre-treatment and NK cell-based therapies.

## 4. Materials and Methods

### 4.1. Isolation and Cultivation of Primary Glioblastoma Stem Cells

GBM tissue samples were obtained from the department of neurosurgery and epilepsy surgery of the Evangelisches Klinikum Bethel, kindly provided by the Forschungsverbund BioMedizin Bielefeld/OWL (FBMB e.V.), Maraweg 21, 33,617 Bielefeld. All experiments involved the patient’s informed consent and were approved by the ethics commission of the University of Münster, Germany, and the General Medical Council at Münster, Germany (approval reference number 2017-522-f-S). GSC isolation was previously described. Briefly, GBM samples were used to isolate primary GSCs by mechanical disintegration, followed by enzymatic digestion with collagenase A (AOF, Worthington, OH, USA) and calcium chloride (300 µM; Sigma-Aldrich, Taufkirchen, Germany). Tissues were centrifuged and cultured in a humidified incubator with 37 °C and 5% CO_2_ in specified selective CSC media containing Dulbecco’s modified Eagle’s medium/F-12 (Life Technologies, Darmstadt, Germany), L-glutamine (2 mM; PAA Laboratories, Linz, Austria), penicillin/streptomycin (100 µg/mL; PAA Laboratories), B-27 (1:50; Gibco, New York, NY, USA), b-FGF (40 ng/mL; Sigma-Aldrich), and EGF (20 ng/mL, Sigma-Aldrich) supplemented with 10% fetal calf serum (FCS; VWR, Darmstadt, Germany) for adherent cultivation. The final enrichment of GSC populations was performed via differential trypsinization, as described previously [42,43]. Here, initially cultured cancer cells were washed with PBS, followed by a 5 min treatment with trypsin (Sigma-Aldrich) and cultured in a new tissue flask. This procedure was performed at least three times to receive stem-like properties in adherently grown GSCs.

### 4.2. Sphere Formation Assay

As an indication of stemness properties in primary GSC populations, isolated cells were seeded in growth media under serum-free conditions. After several days of culture, exemplary images of formed spheres were performed by using light microscopy (EVOS XL; Thermo Scientific, Waltham, MA, USA).

### 4.3. Immunochemistry

To stain GBM slices immunohistochemically, paraffin-embedded sections were deparaffinized and rehydrated first. GBM slices were washed with xylol for 20 min, followed by 10 min in ethanol absolute, and three rehydration steps for 5 min by using 90%-, 80%, and 70% ethanol. Reconditioning of the epitope was performed with boiling of GBM sections for 20 min in 0.01 M citrate buffer (pH 6.0) and chilled afterward for 30 min at room temperature. Next, slices were washed in PBS with 0.02% Triton-X 100 (Sigma-Aldrich) for 10 min. Free binding sides were blocked and permeabilizated via incubation with 0.02% Triton-X 100, 10% appropriative serum (DIANOVA, Hamburg, Germany), and 1% bovine serum albumin (Sigma-Aldrich) in PBS for at least 2 h. Afterward, primary antibodies were diluted in blocking solution and applied for 1 h: anti-CD44 (1:50; 156-3C11; Cell Signaling, Frankfurt am Main, Germany) and anti-CD133 (1:100; NB120-16,518; NovusBio, Bio-Techne, Wiesbaden-Nordenstadt, Germany). GBM slices were washed with PBS, followed by incubation with the respective secondary fluorochrome-conjugated antibodies (Alexa Flour 555 and -488 dyes, 1:300; goat anti-mouse and goat anti-rabbit; Life Technologies) for 1 h under the exclusion from light. Subsequent incubation with 4′,6-diamidino-2-phenylindole (DAPI; 1 μg/mL; Sigma-Aldrich) was followed by nuclear counterstaining for 10 min. Fluorescence imaging was performed via a confocal laser scanning microscope (LSM 780; Carl Zeiss, Jena, Germany) and analyzed via ImageJ software (NIH, Bethesda, MD, USA).

For immunocytochemical staining of adherent cells, GSCs were seeded at the top of sterilized coverslips with 50,000 cells per 4 cm^2^ in 24-well plates with regular growth medium and cultured initially for 48 h. Fixation was performed with 4% paraformaldehyde in PBS for 10 min. Free binding sides of GSCs were blocked as well as permeabilization was processed by the usage of PBT solution, including 0.02% Triton-X-100 and 5% goat serum (DIANOVA) in PBS for 30 min. Three washing steps with PBS followed as well as an 1 h incubation with the primary antibodies: anti-CD44 (1:400), anti-CD133 (1:100), anti-Nestin (1:200; MAB5326; Merck), anti-NF-κB p65 (RelA; 1:400; D14E12; Cell Signaling), anti-RelB (1:100; D7D7W; Cell Signaling), anti-cRel (1:100; Cell Signaling), anti-MYC (10 μg/mL; Y69; Abcam) and anti-N-MYC (2.5 μg/mL; NCM II 100; Abcam). After further washing steps with PBS, secondary fluorochrome-conjugated antibodies (Alexa Fluor dyes 555 and -488; 1:300) were applied for 1 h, protected from light. Nuclear staining was performed using DAPI (1 µg/mL) for 10 min. Before GSCs were embedded in Mowiol 4–88 (Carl Roth GmbH, Karlsruhe, Germany) upside down on the top of microscope slides, a final washing step with distilled water was performed. For fluorescence imaging, the confocal laser-scanning microscope LSM 780 was used. Protein localizations were analyzed similarly to immunohistochemical sections.

Immunostainings of GSC spheres were processed with fixation in 4% paraformaldehyde for 2 h; fixed free-floating cells were further embedded in paraffin and were sectionalized afterward. Preparation of sphere slices was performed equally to immunohistochemical stainings of GBM sections. Primary antibodies used for spheroid stainings: anti-CD44 (1:50), anti-CD133 (1:100), anti-Nestin (1:200), anti-MYC (10 μg/mL), and anti-NF-κB p65 (1:800). Slices were washed three times with PBS, and another 1 h incubation with Alexa Fluor 555 and -488 secondary fluorochrome-conjugated antibodies (1:300) was performed. Nuclear staining by using DAPI (1 µg/mL) as well as fluorescence imaging was processed similarly to immunohisto- and immunocytochemistry.

### 4.4. RNA Isolation and Non-Quantitative/Quantitative Reverse-Transcription PCR

RNA from 1 × 10^6^ cultured and pooled GSCs were isolated using the NucleoSpin^®^ RNA Plus kit (Macherey-Nagel, Düren, Germany) according to the manufacturer’s instructions. The quality and concentration of isolated RNAs were assessed via nanodrop ultraviolet spectrophotometry. For each sample, 1 µg of RNA was transcribed into copy DNA (cDNA) via the First Strand cDNA Synthesis Kit (Thermo Scientific) using the related random hexamer primers. GSC gene expression profiles were analyzed using recombinant taq-polymerase (5 U/µL) and the associated protocol according to Thermo Scientific’s guidelines. Target sequences used for characterization are listed below (Table 2).

Amplified transcripts were loaded on agarose gels (Sigma-Aldrich) with 0.001% ethidium bromide (Carl Roth GmbH), ran by 100 V, and were subsequently imaged with a trans-illuminator (UVsolo TS; Biometra, Göttingen, Germany). *B-ACTIN* served as a housekeeping gene.

For quantitative RT-PCR, regular pre-cultured GSCs were partially treated with TNFα (10 ng/mL) and/or PDTC (100 µM; Peprotech) for 3 h, and PDTC and/or KJ-Pyr-9 and DMSO as solvent control for 24 h. RNA isolation, as well as cDNA synthesis, were processed as described above. Synthesized cDNAs were analyzed via quantitative RT-PCRs using the PerfeCTa SYBR Green SuperMix as well as the Rotor Gene 6000 cycler, according to the manufacturer’s protocols. Quantitative expression analysis was performed in triplicate with the following sequences (Table 3):

Data evaluation of quantitative RT-PCR results was performed via the LinRegPCR program [84], whereas *B-ACTIN* and *GAPDH* served as housekeeping genes for normalization.

### 4.5. Natural Killer Cell Isolation and Culture

NK cells from primary human buffy coats (kindly provided by the Institut für Laboratoriums und Transfusionsmedizin, HDZ NRW, Bad Oeynhausen, Germany) were enriched via density gradient centrifugation, using SepMate™-50 tubes, Lymphoprep™, and the RosetteSep™ human NK cell enrichment cocktail and remaining red blood cells were lysed using the EasySep™ Red Blood Cell Lysis Buffer (all STEMCELL Technologies Inc., Vancouver, BC, Canada), all according to the manufacturer’s instructions. The isolated population consisted of 97% NK cells (88.6% CD56 dim and 8.5% CD56 bright) and less than 0.4% contaminating T and NKT cells. The cells were cultured in NK MACS^®^ Medium (Miltenyi Biotec, Bergisch Gladbach, Germany) supplemented with penicillin/streptomycin (100 µg/mL; PAA Laboratories), 10% human AB serum (PAN-Biotec, Aidenbach, Germany) and 500 U/mL Human IL-2 IS Premium (Miltenyi Biotec, Bergisch Gladbach, Germany). Fresh Media was added every three to four days, and the cells were passaged once they reached 80% confluency. Cells were harvested via trypsin and used in flow cytometry-based cytotoxicity assays.

### 4.6. Flow Cytometry

To determine the NK cell purity after isolation, the cells were stained with DAPI (Sigma-Aldrich), CD3-FITC (clone OKT3; BioLegend, San Diego, CA, USA), and CD56-APC-A700 (clone N901; Beckman Coulter, Brea, CA, USA). Stained cells were measured by flow cytometry using a Gallios Flow Cytometer (Three Laser, 10 Color Filter Block Configuration) from Beckman Coulter Life Sciences (Krefeld, Germany). Analysis was performed with the Kaluza 1.30 software (Beckman Coulter Life Sciences), and compensation was performed using the integrated automatic compensation tool with appropriate single stainings. In analysis, a Side-Scatter-Area vs. time plot was used to gate for areas with undisturbed flow. Dead cells were removed on a Side-Scatter-Area vs. DAPI (405 nm excitation, FL9 detection) plot. Singlets were selected on a Forward-Scatter-Height vs. Forward-Scatter-Area plot, and debris was removed by plotting the Side-Scatter-Area vs. the Forward-Scatter-Area, drawing gates around all cells and lymphocytes. The lymphocytes were visualized on a CD56-APC-A700 (633 nm excitation, FL7 detection) vs. CD3-FITC (633 nm excitation, FL1 detection) plot, and regions were drawn around CD56 dim and CD56 bright NK cells.

To measure the cytotoxic activity of NK cells against GSCs, GSCs were cultured adherently in regular growth media as described above and were harvested via trypsin. Cells were stained with 1:2000 diluted Cell Trace Violet (Thermo Fisher Scientific, Waltham, MA, USA) for 20 min at 37 °C and washed twice in PBS. In a standard target cell culture medium, stained cells were incubated with NK cells in different ratios (1:0, 1:1, 1:3, 1:9 target to effector cells) for four hours, each in triplicates. Dead cells were stained using Propidium Iodide. All samples were measured on a Gallios Flow Cytometer (Three Laser, 10 Color Filter Block Configuration) from Beckman Coulter Life Sciences (Krefeld, Germany), and obtained data were analyzed using the Kaluza 1.30 software (Beckman Coulter Life Sciences). A Side-Scatter-Area vs. time plot was used to gate for areas with undisturbed flow, and debris was removed by plotting the Side-Scatter-Area vs. the Forward-Scatter-Area. Target cells were selected on a Side-Scatter-Area vs. Cell Trace Violet (405 nm excitation, FL9 detection) plot. On a Side-Scatter-Area vs. Propidium Iodide (488 nm excitation, FL3 detection) plot, the proportion of dead to living target cells was determined. Specific lysis was calculated as follows:(1)$$specific\;lysis=\frac{living\;target\;cells\;control\;\left[\%\right]\;-\;living\;target\;cells\;test\;\left[\%\right]}{living\;target\;cells\;test\;\left[\%\right]}\times 100$$

To determine the ALDH1 activities in primary GSC populations, cells were initially cultured adherently in regular growth media as described above. Afterward, cells were harvested via trypsin and analyzed by flow cytometry on a Gallios Flow Cytometer (Three Laser, 10 Color Filter Block Configuration, Beckman Coulter Life Sciences). The measurement of the ALDH1^high^ activity was performed by using the ALDEFLUOR™ Kit (STEMCELL Technologies Inc., Vancouver, BC, Canada) according to the manufacturer’s instructions. Dead cells were excluded via Propidium Iodide (Sigma-Aldrich), and automatic compensation values were determined using the automated compensation tool of the Kaluza 1.3 software (Beckman Coulter Life Sciences) with appropriate single stainings. The results were analyzed using FlowJo™_v10.8.1 Software for Windows (Becton, Dickinson and Company, Ashland, OR, USA). Briefly, the FlowAI Plugin in default setting was used to clean the data. Debris was removed by plotting the Side-Scatter-Area vs. the Forward-Scatter-Area, and singlets were selected on a Forward-Scatter-Height vs. Forward-Scatter-Area plot. Dead cells were removed on a Side-Scatter-Area vs. Propidium Iodide (488 nm excitation, FL3 detection) plot. Subsequently, the DownSampleV3 Plugin was used to obtain even event counts between the samples and corresponding controls. The data were visualized in a Side-Scatter-Area vs. Aldefluor (488 nm excitation, FL1 detection) plot, and gates were drawn according to the corresponding controls with a tolerance of 0.5% false positive or false bright events, respectively.

### 4.7. Activation of the NF-κB Transcription Factor

For a potential stimulation of the NF-κB transcription factor, GSC populations were initially seeded on the top of sterilized coverslips in 24-well plates with regular growth media as described for immunocytochemical stainings. After 48 h of culture, the medium was refreshed and supplemented with the cytokine TNFα (10 ng/mL; Merck). Cells were stimulated for 10, 30, 60, and 90 min to analyze unique effects on nuclear RELA and c-REL protein in GSCs. Fixation and immunofluorescent stainings were performed as mentioned above. Here, at least five images were analyzed per condition and target protein. Fold changes of nuclear RELA, as well as nuclear c-REL intensities, were examined via the following formula:(2)$$Fold\;change=\frac{\left(F_{treatment}-F_{min}\right)}{\left(F_{max}-F_{min}\right)}\times 100$$

### 4.8. Haploid Copy Number Alteration

Genomic DNA isolation for determining the gene copy number of *MYC* and *MYCN* was performed using the QUIamp^®^ DNA Mini Kit (Qiagen). Subsequent quantification was processed via q-RT-PCR in triplicates, using the PerfeCTa SYBR Green SuperMix (QuantaBio, Beverly, USA) and the Rotor Gene 6000 (Qiagen), according to the manufacturer’s instructions. For each amplified gene, 10 ng of human genomic calibrator DNA (Roche Diagnostics, Mannheim, Germany) and 10 ng of isolated genomic GSC DNA and a non-template control were used. The respective haploid copy number was examined by using the previously described equation by De Preter et al. [85]:(3)$$2^{-\Delta \Delta C_{T}}=\frac{{\left(1+E\right)}^{-\Delta C_{T\;gene}}}{{\left(1+E\right)}^{-\Delta C_{T\;reference}}}$$

To detect *MYC* and *MYCN* gene copy numbers, specific genomic primer sequences were used and listed below next to *GAPDH* and *SYNDECAN4*, which were utilized as reference genes (Table 4).

### 4.9. Migration Assay

To investigate the migrative potential and the subsequent protein expression profiles of cultured GSC spheroids, primary cells were firstly cultured in the absence of FCS, as described in Section 4.2. After formation, GSC spheres were placed in 8-well chambered coverslips (laser microscopy-suitable bottom; ibidi GmbH, Gräfelfing, Germany) with fresh CSC media, including 10% FCS, and cultured for 72 h. By the addition of serum, primary spheres deposited and migrated out adherently. Further immunocytochemical staining of MYC and RELA protein was performed directly within the 8-well chambers, similar to 4.3. Migration zones and specific protein localizations were processed and analyzed via confocal laser-scanning microscope LSM 780 following ImageJ software.

### 4.10. Inhibitor Treatments and CompuSyn Analysis

To examine GSC cellular viabilities after treatment with TNFα, PDTC, TMZ, and/or the specific MYC inhibitor KJ-Pyr-9, 3 × 5000 cultured cells for each population were seeded per 0.4 cm^2^ in 96-well plates and incubated initially for 2 h in regular growth media. After GSC attachment, cells were treated with PDTC in concentrations ranging from 0.5 to 100 µM and KJ-Pyr-9 in concentrations ranging from 1 to 60 µM. Additionally, for the combination assay, cells were treated either alone or in combination with TNFα (10 ng/mL), PDTC (100 µM), TMZ (250 µM; Sigma-Aldrich), and/or KJ-Pyr-9 (40 µM; Merck). GSCs were cultured up to a final duration of 96 h with an additional charge of the treatment media after 48 h. To determine the metabolic rates displayed as the specific absorptions at 450 nm, the Orangu™ Cell Counting Solution (Cell Guidance Systems Ltd., Cambridge, UK) was used according to the manufacturer’s instructions. Calculation of the subsequent cellular viabilities of GSCs were performed as previously described [86] and by using the following formula:(4)$$Cellular\;viability=\frac{\left(abs_{treatment}\;-\;abs_{0}\right)}{\left(abs_{control}-abs_{0}\right)}\times 100\%$$

The half maximal inhibitory concentration (IC_50_) was calculated from a log (concentration) versus normalized viability rate non-linear regression fit using Prism V5.01 software (GraphPad Software, Inc., San Diego, CA, USA). Actions between PDTC and KJ-Pyr-9 were analyzed using CompuSyn (ComboSyn Inc, Paramus, NJ, USA). Here, single-drug dosis-effect values, and combined dosis-effect values, were used to determine CI values and display the normalized non-constant ratio isobolograms, according to Chou [87].

### 4.11. Statistical Analysis and Figure Design

Data were analyzed using the Prism V5.01 software (GraphPad Software, Inc., San Diego, CA, USA). Tests for normality were performed via D’Agnostino & Pearson omnibus normality test. Non-parametric one-way ANOVA (Kruskal-Wallis test) with Dunn’s multiple comparison test, Mann-Whitney U test, or unpaired t-tests were conducted to assess differences between multiple groups. For all analyses, a *p* ≤ 0.05 value was considered statistically significant. Evaluated data are displayed as the means ± SEM. All figures were designed via the CorelDRAW Graphics Suite 2018 software (Corel Corporation, Ottawa, ON, Canada).

## Figures and Tables

**Figure 1 ijms-23-12919-f001:**
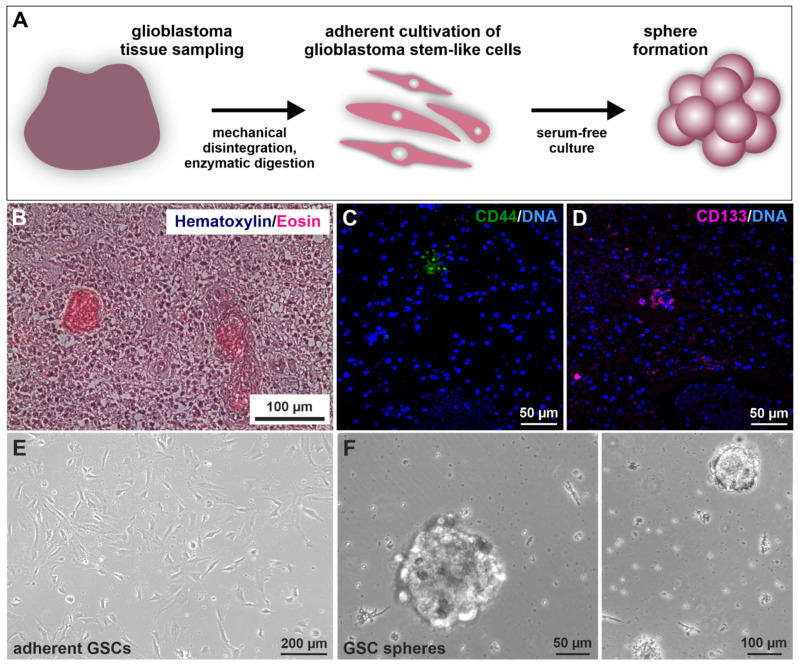
Isolation and cultivation of primary glioblastoma stem cells (GSCs). (**A**) Schematic representation of the practical procedure after surgical removal of the tumor. Tissues were sampled, minced, digested, and afterward cultivated under specified selecting/enriching conditions. (**B**) Exemplary histopathological hematoxylin/eosin staining showed typical cancer cell accumulations within the analyzed GII tumor tissue. Immunostaining of exemplary stained GIV tissue samples for the common cancer stem cell markers (**C**) CD44 and (**D**) CD133 depicted specific protein expressions localized in small subgroups of clustered cells. 4′,6-diamidino-2-phenylindol (DAPI) served for nuclear counterstaining. Following differential trypsinization, mesenchymal-shaped cells grew (**E**) adherently with fetal calf serum and were also able to (**F**) form spheres without the addition of serum.

**Figure 2 ijms-23-12919-f002:**
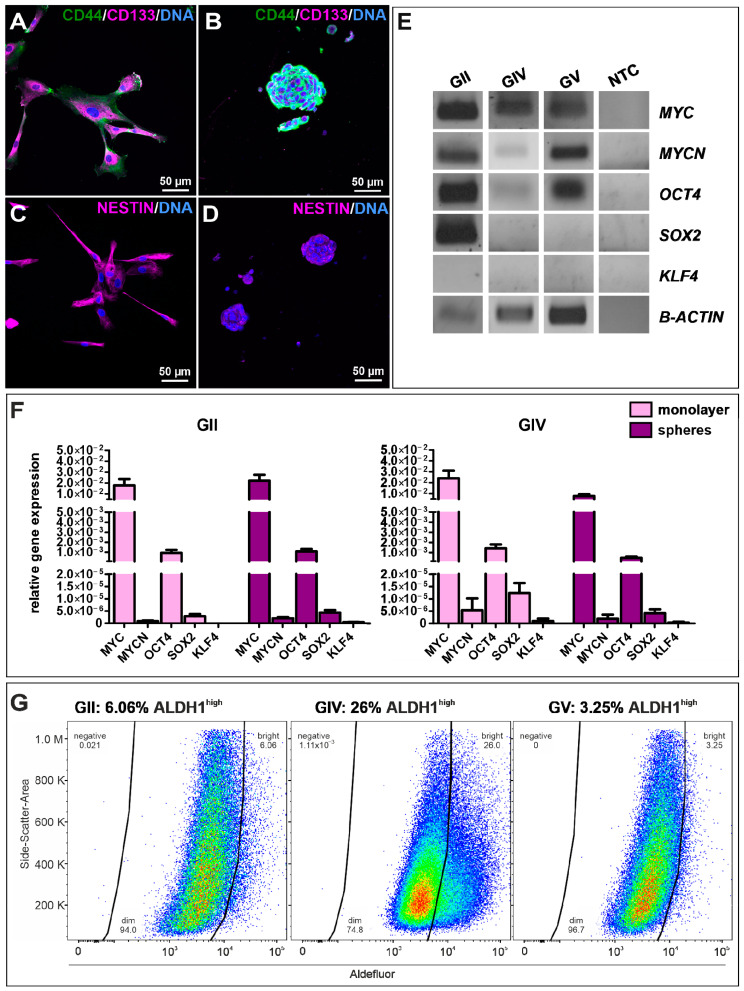
Characterization and identification of stemness properties in primary GSCs. Immunocytochemical stainings of the CSC markers CD44 (green) and CD133 (magenta) in GSCs, after (**A**) adherent growth (exemplary GSC population from donor GII) and in (**B**) cultured spheres (exemplary (GSC population from donor GV). Presence of Nestin further indicated the stemness character of isolated GSCs, which were also detected via immunocytochemistry of (**C**) adherent GSCs (exemplary GSC population from donor GII) and after (**D**) sphere formation (exemplary GSC population from donor GV). DAPI served as nuclear counterstaining. (**E**) Qualtitative reverse transcriptase polymerase chain reaction (RT-PCR) of *MYC*, *MYCN*, *OCT4*, *SOX2,* and *KLF4* in adherent GSC shows similar expression levels. (**F**) Quantitative RT-PCR of *MYC*, *MYCN*, *OCT4*, *SOX2,* and *KLF4* in adherent GSCs and spheres reveals similar expression levels in GII and GIV. (**G**) Flow cytometry of ALDH1 activities in primary GSCs showed respective ALDH1^high^ amounts of 6.06% for GII, 26.00% for GIV, and 3.25% for GV.

**Figure 3 ijms-23-12919-f003:**
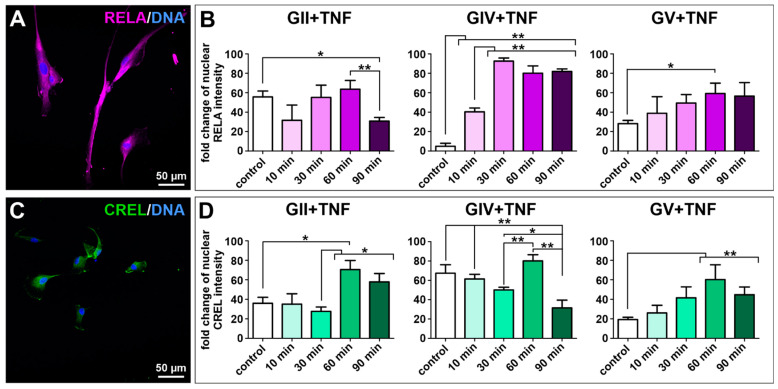
Tumor necrosis factor α-induced activation of the NF-κB transcription factor in primary GSCs. Confocal analysis revealed (**A**) expression of NF-κB subunit RELA (exemplary GSC population from donor GII). (**B**) Quantifying nuclear RELA intensity measured by confocal analysis showed a significant increase of the fold change after stimulation with TNFα for GIV GSCs after 10 min and GV GSCs after 60 min. (**C**) GCSs express NF-κB subunit c-REL (exemplary GSC population from donor GII). (**D**) Quantification of nuclear c-REL intensity showed a significant increase of the fold change after stimulation with TNFα for GII GSCs, GIV GSCs as well as GV GSCs after 60 min. Means ± SEM (standard error of the mean) were statistically analyzed by Mann Whitney U tests (*n* = 5, * *p* ≤ 0.05, ** *p* ≤ 0.01).

**Figure 4 ijms-23-12919-f004:**
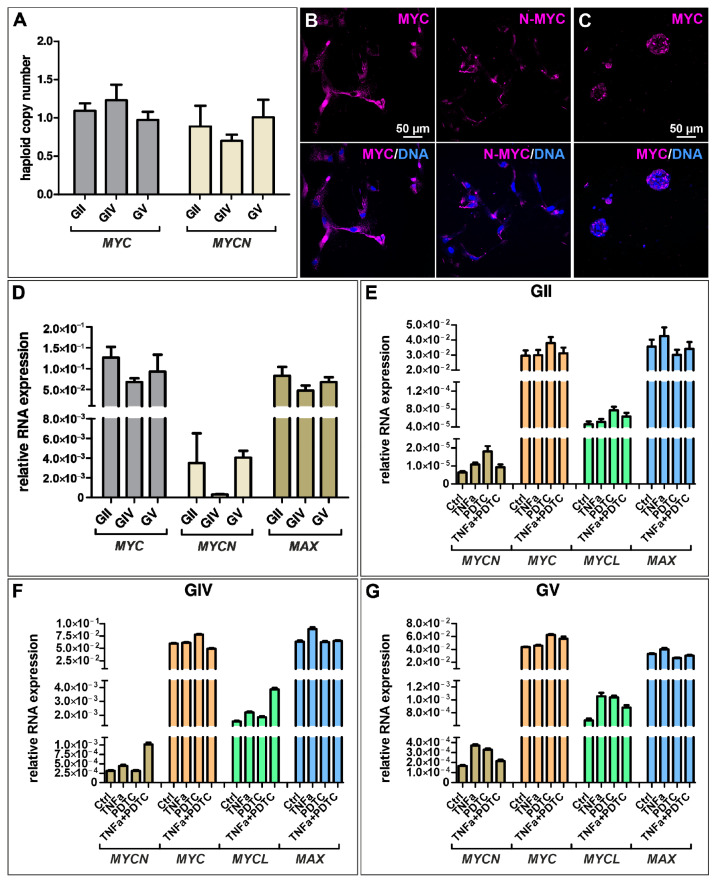
Expression of MYC-family members is present in GSCs but not regulated through NF-κB. (**A**) Copy number alteration assay reveals a normal copy number of *MYC* and *MYCN* in primary GSCs. Immunocytochemical staining revealed robust expression of (**B**) *MYC* and *N-MYC* in adherently growing GSCs (exemplary GSC population from donor GV) as well as (**C**) *MYC* in GSC-derived spheres (exemplary GSC population from donor GV). (**D**) Quantitative mRNA expression analysis showed a higher expression of *MYC* and *Myc-associated factor X* (*MAX)* than *MYCN*. (**E**–**G**) Treatment with TNFα and/or NF-κB inhibitor pyrrolidine dithiocarbamate (PDTC) did not significantly change the expression levels of *MYCN*, *MYC*, *MYCL,* and *MAX* in GSCs.

**Figure 5 ijms-23-12919-f005:**
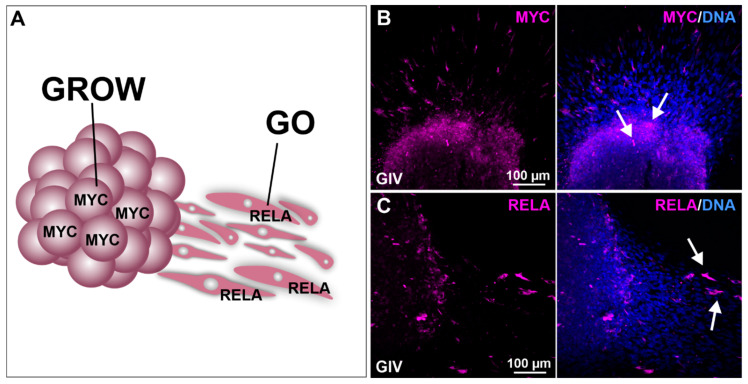
Migration analysis shows involvement of MYC in cell growth and RELA in the migration of GSCs. (**A**) Schematic representation of MYC and RELA involvement in cellular growth (GROW) and migration (GO). Deposited spheres show (**B**) high MYC expression within the sphere as well as (**C**) high RelA expression in GSCs migrating out of the sphere (white arrows).

**Figure 6 ijms-23-12919-f006:**
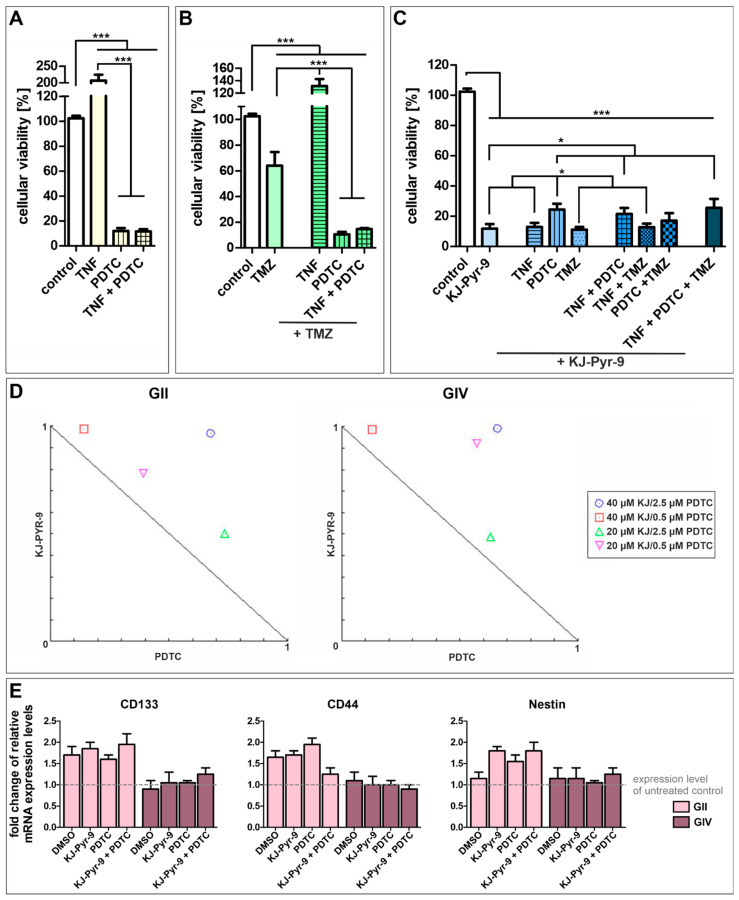
High-level reduction of GSC viability via NF-κB and MYC inhibition. Pooled cellular viabilities of GSC populations could be reduced via treatment with (**A**) 100 µM PDTC, reversing the cytoprotective effect of TNFα. (**B**) Using 250 µM temozolomide (TMZ) decreased the cell viability of GCSs, while PDTC enhanced the cytotoxic effect. Combination of TMZ with TNFα led to cytoprotective effects. (**C**) Treatment with 40 µM of the MYC inhibitor KJ-Pyr-9 resulted in significantly decreased cellular viabilities alone or in combination with TNFα, TMZ, and PDTC. (**D**) Normalized isobolograms show antagonistic actions of KJ-Pyr-9 and PDTC in GII and GIV GSCs. (**E**) Quantitative RT-PCR revealed a higher expression of CD133, CD44, and Nestin in GII GSCs after treatment and similar expression levels of CD133, CD44, and Nestin in GIV GSCs compared to untreated control. Means ± SEM (standard error of the mean) were statistically analyzed by Mann Whitney U tests (*n* = 9, * *p* ≤ 0.05, *** *p* ≤ 0.001).

**Figure 7 ijms-23-12919-f007:**
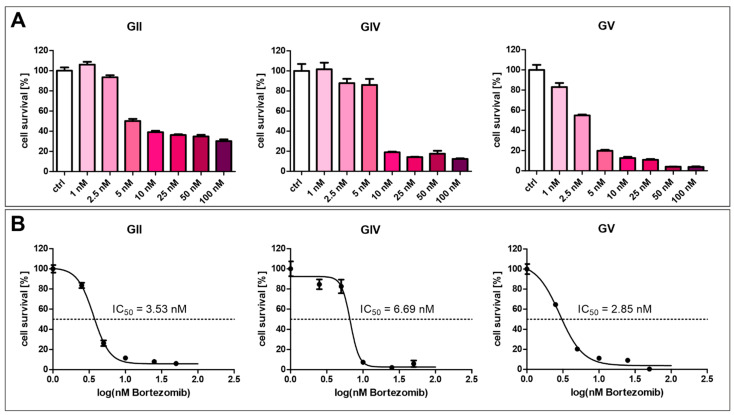
Bortezomib-mediated inhibition of GSC survival. (**A**) Treatment of GCSs with Bortezomib in concentrations higher than 2.5 nM inhibited cell survival. (**B**) Calculating half maximal inhibitory concentration revealed values between 2.85 and 6.69 nM.

**Figure 8 ijms-23-12919-f008:**
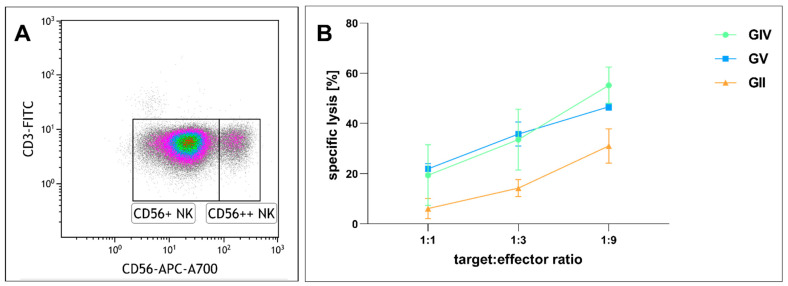
Cytotoxicity of cultured human natural killer (NK) cells against GSCs. (**A**) NK cell population after isolation. Lymphocytes plotted as CD3 vs. CD56 expression. CD56 dim (CD56+) NK cells equaling 88.6% of total cells and CD56 bright (CD56++) NK cells 8.5% of total cells. (**B**) Human NK cells cultured with 500 U/mL IL2 for 13, 20, and 26 days were co-incubated with GSCs at different ratios for four hours. Depicted is the specific lysis of GSCs as means ± SD (standard deviation). Statistical analysis with one-way ANOVA detected no significant differences between the different GSC populations.

**Figure 9 ijms-23-12919-f009:**
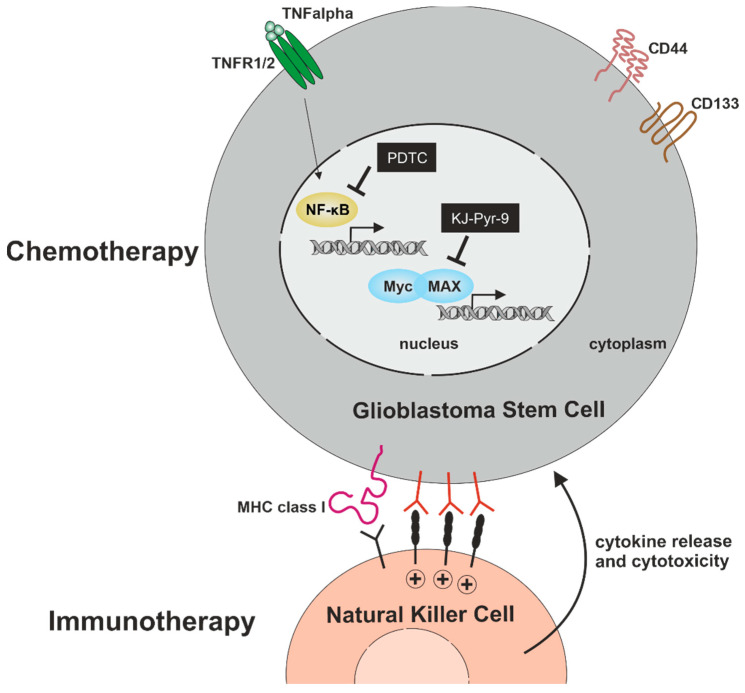
Targeting GSCs for the development of efficient chemo- and immunotherapy. Pharmacologic inhibition of NF-κB by PTDC or inhibition of MYC by KJ-Pyr-9 significantly reduces GSC-viability with no cytoprotective effect of TNFα. As an additional cell-therapeutic strategy, NK cells are capable of killing GSCs.

**Table 1 ijms-23-12919-t001:** Classification of patients’ donated glioblastoma multiforme samples.

Donor ID	Tumor Typing and Characterization	WHO Grade	Sex	Age
GII	Primary glioblastoma multiforme, *IDH1* wildtype with *MGMT* promoter methylation	IV	Female	86
GIV	Secondary glioblastoma multiforme, *IDH1* mutation with *MGMT* promoter methylation	IV	Male	60
GV	Primary glioblastoma multiforme, *IDH1* wildtype without *MGMT* promoter methylation	IV	Male	42

**Table 2 ijms-23-12919-t002:** Primer sequences used for gene amplification via polymerase chain reaction.

Target Gene	Sequence (5′-3′)
*B-ACTIN*	GAGAAGATGACCCAGATCATGT
	CATCTCTTGCTCGAAGTCCAG
*KLF4*	CAGCTTCACCTATCCGATCC
	TGTACACCGGGTCCAATTCT
*MYC*	GGCACTTTGCACTGGAACTT
	AGGCTGCTGGTTTTCCACTA
*MYCN*	ACAGTCATCTGTCTGGACGC
	TCCTCGGATGGCTACAGTCT
*OCT4*	CGAAAGAGAAAGCGAACCAG
	GCCGGTTACAGAACCACACT
*SOX2*	GGAGCTTTGCAGGAAGTTTG
	GCAAGAAGCCTCTCCTTGAA

**Table 3 ijms-23-12919-t003:** Primer sequences used for quantitative polymerase chain reaction.

Target Gene	Sequence (5′-3′)
*B-ACTIN*	CTTCGCGGGCGACGAT
	CCACATAGGAATCCTTCTGACC
*GAPDH*	CATGAGAAGTATGACAACAGCCT
	AGTCCTTCCACGATACCAAAGT
*SOX2*	GGAGCTTTGCAGGAAGTTTG
	GCAAGAAGCCTCTCCTTGAA
*KLF4*	CAGCTTCACCTATCCGATCC
	TGTACACCGGGTCCAATTCT
*OCT4*	CGAAAGAGAAAGCGAACCAG
	GCCGGTTACAGAACCACACT
*MYC*	GGCACTTTGCACTGGAACTT
	AGGCTGCTGGTTTTCCACTA
*MYCN*	ACAGTCATCTGTCTGGACGC
	TCCTCGGATGGCTACAGTCT
*MAX*	AGGCTGCTGGTTTTCCACTA
	TGAGTCCCGCAAACTGTGAA
*MYCL*	ACCAGCTGTCTTGGGTGAAG
	TTAAGTGTTCCCAGGGTCGC
*CD133*	AACAGTTTGCCCCCAGGAAA
	GAAGGACTCGTTGCTGGTGA
*CD44*	CTACAAGCACAATCCAGGCA
	GCATTGGATGGCTGGTATGA
*NESTIN*	CGCACCTCAAGATGTCCCTC
	CAGCTTGGGTCCTGAAAGC

**Table 4 ijms-23-12919-t004:** Primer sequences used for genomic real-time polymerase chain reaction.

Target Gene	Sequence (5′-3′)
*GAPDH*	AGACTGGCTCTTAAAAAGTGCAGG
	TGCTGTAGCCAAATTCGTTGTC
*MYC*	AAAAGTGGGCGGCTGGATAC
	AGGGATGGGAGGAAACGCTA
*MYCN*	CGCAAAAGCCACCTCTCATTA
	TCCAGCAGATGCCACATAAGG
*SYNDECAN4*	CAGGGTCTGGGAGCCAAGT
	GCACAGTGCTGGACATTGACA

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
