# Peer review of "Targeting Key Signaling Pathways in Glioblastoma Stem Cells for the Development of Efficient Chemo- and Immunotherapy"

_ijms, 2022, doi:10.3390/ijms232112919_

Round 1
Reviewer 1 Report (New Reviewer)
The authors, Helweg and colleagues, inhibited crucial signaling pathways in glioblastoma stem cells (GSC) to impair the cell viability of GSC in situ. Furthermore, it was shown that an NK-cell activation leads to the destruction of GSC. The manuscript is well-written and clear, and the experiences performed are solid. However, the manuscript lacks novelty since the inhibitors used in the assays are well characterized and the outcome of the NK-cell assay is well known.
Minor concerns
1. Line 73: The activation receptor NKG2D should be described.
2. Results, figure 6: It might be noteworthy to add TGF-beta to the assay.
Sincerely,
Author Response
Please see the attachment.

Reviewer 2 Report (New Reviewer)
In this manuscript, the authors employed 3 glioblastoma stem cells they built before to test the probable therapeutic method. Based on the high expression of NFKB, there is pharmacologic inhibition of NF-κB and MYC by the specific compounds. This could effectively inhibit tumor growth.
Comments on this manuscript,
1. CD133 is a cell surface marker, and why it looks located in the cytoplasm in fig2 A,C.
2. Is it the immunohistochemistry or immunofluorescence in fig1 C and D, and why did you choose the GII tumor for HE but GIV tumor for the CD44 and CD133 markers detection?
3. There were several black dots in the background of Fig1 E and F, are they contaminated?
4. No in vivo data support the conclusion of this manuscript.
5. Are there any normal cells for the control of all study, like the CD44/133 expression, cell viability? In the same concentration of these compounds, do they have selectivity for the tumor cells?
6. “Human NK cells cultured with 500 U/ml IL2 for several days”, how many days are called several?
7. In fig 2E why not include G V?
8. Why did you evaluate the NK cells for the cell-mediated lysis of glioblastoma stem cells, not the CD8 T cells or macrophage?
9. Did these inhibitors penetrate the blood brain barrier?
10. Some small grammar errors need to be checked.
Round 2
Reviewer 1 Report (New Reviewer)
Comments sent to the editor.
Reviewer 2 Report (New Reviewer)
The authors addressed my concerns about the manuscript, from my perspective, I have no other questions.
This manuscript is a resubmission of an earlier submission. The following is a list of the peer review reports and author responses from that submission.
Round 1
Reviewer 2 Report
Dear All,
Re: Targeting key signaling pathways in glioblastoma stem cells for the development of efficient chemo- and immunotherapy.
The manuscript's authors investigated NF-kappa-B signaling in glioma stem cells (GSCs). Three different patients' GSCs were analyzed, showing that TNF-alpha activated NF-kappa-B-RELA and NF-kappa-B-c-Rel depending on the GSC type from each patient. Moreover, their investigations revealed that MYC was more expressed in the tumorspheres. By contrast, NF-kappa-B-RELA was more expressed in the migrating GSCs. Furthermore, the pharmacological targeting to inhibit MYC or NF-kappa-B led to the loss of GSC viability. Finally, the authors demonstrated that NK cells could kill GSCs.
1) Figure 2 needs clarification. In figure 2E, NTC is not defined. NTC needs to be defined.
2) Figures 6A, B, and C do not refer to which GSC type for which the data were obtained. Therefore, figure 6 needs to be clarified which patient type was or was used for 6A, B, and C.
3) In Figure 7, only GIV GSCs was used to demonstrate that NK cells can kill GSCs. Therefore, the experiment should be repeated with an additional GSC type for reproducibility.